# Assessment of Mental Health Factors among Health Professionals Depending on Their Contact with COVID-19 Patients

**DOI:** 10.3390/ijerph17165849

**Published:** 2020-08-12

**Authors:** Paweł Wańkowicz, Aleksandra Szylińska, Iwona Rotter

**Affiliations:** Department of Medical Rehabilitation and Clinical Physiotherapy, Pomeranian Medical University in Szczecin, 71-210 Szczecin, Poland; aleksandra.szylinska@gmail.com (A.S.); iwrot@wp.pl (I.R.)

**Keywords:** SARS-CoV-2, COVID-19, Generalized Anxiety Disorder scale (GAD-7), Patient Health Questionnaire (PHQ), Insomnia Severity Index (ISI)

## Abstract

It seems that the medical personnel in contact with patients infected with SARS-CoV-2 are at an especially high risk of adverse psychological effects. Therefore, the aim of this study was to assess the mental health factors among healthcare workers by quantifying the severity of anxiety, depression, and sleep disorders during the current SARS-CoV-2 pandemic, while taking into account coexisting diseases. The study involved 441 healthcare professionals including 206 healthcare workers at emergency wards, infectious wards, and intensive care units. The control group consisted of 235 healthcare workers working in wards other than those where individuals from the study group worked. Regression adjusted by age, gender, the occurrence of hypertension, diabetes mellitus, dyslipidemia, asthma, autoimmune diseases, and cigarette smoking showed the elevated risk of anxiety on the Generalized Anxiety Disorder (GAD-7) scale (OR = 1.934; *p* < 0.001), depression on the Patient Health Questionnaire (PHQ-9) scale (OR = 2.623; *p* < 0.001), and sleep disorders on the Insomnia Severity Index (ISI) scale (OR = 3.078; *p* < 0.001). Our study showed that healthcare workers who are exposed to SARS-CoV-2-infected patients at emergency wards, infectious wards, and intensive care units are at a much higher risk of showing symptoms of anxiety, depression, and sleep disorders than healthcare workers working in other wards.

## 1. Introduction

In December 2019, a new type of coronavirus was identified in Wuhan, China, following a growing number of reports of new atypical pneumonia [1]. The virus, named SARS-CoV-2 by the International Committee on Virus Taxonomy due to its similarity to the severe acute respiratory syndrome coronavirus [2], causes a disease known as COVID-19. To date, it has killed 530,000 people and infected more than 11 million, causing a pandemic that poses a serious challenge to healthcare systems around the world [3]. In Poland, there have been 45,000 confirmed cases and 1700 deaths.

In recent years, we have seen the unprecedented acceleration of the transmission of various viral infections as a result of climate change, the speed of population movement, the appropriation of pristine forests, and the reduction in the distance to the species that live there, including microorganisms. Infectious diseases have accompanied humanity for centuries [4], always triggering various psychological reactions and changing the behavior patterns of entire communities. The lockdown introduced by the Polish government on 15 March 2020 dramatically affected the daily life of the society. Isolation and many other imposed limitations may increase psychological distress, including the symptoms of depression and anxiety, as confirmed by research conducted around the world [5,6]. Although this response of the human psyche to threats and uncertainty is natural, in some individuals, it may exceed an ability to adapt and cope, leading to the development of clinically significant symptoms. Certain groups of people are particularly vulnerable to stress due to the consequences of the pandemic and, as a result, are more likely to develop depression and generalized anxiety disorder. For some, this may be related problems from before the epidemic, e.g., people with a precarious financial situation may fear that the pandemic will make it even worse; those entering adulthood and still living with their parents may face increased interpersonal conflicts at home. For others, previously natural situations, may develop into difficult situations, increasing anxiety or depression, such as when living alone may lead to the overwhelming sense of loneliness when meeting other becomes impossible [7,8]. Others have to deal with completely new and serious challenges, e.g., students and teachers who switch to online education and fear of not being able to cope, with insufficient resources and competence in the field of new communication technologies [9]. However, the group of people who seem to be most exposed to exceptionally high level of stress during the pandemic are health professionals, burdened with emotionally challenging interactions with the sick and potentially dying persons, fearing for their own and their families’ health, and subject to occupational overload due to staff shortages and insufficient personal protective equipment [10]. These circumstances may increase the risk of mental health disorders, resulting in depression, anxiety, insomnia, and suicide. Even in times before the pandemic, physicians had the highest suicide rate of all professions. The previous surveys of health professionals during the SARS epidemic in 2003 and SARS-CoV-2 pandemic in 2019 found significant levels of anxiety and stress which considerably affect their work and quality of sleep [11,12,13,14,15,16,17]. 

Therefore, this study aimed to assess mental health factors among healthcare workers by quantifying the severity of symptoms of anxiety, depression, and sleep disorders during the current SARS-CoV-2 pandemic, while taking into account coexisting diseases.

## 2. Materials and Methods

This is a cross-sectional, hospital-based study conducted in the Western Pomerania region in Poland from 3 May 2020 to 17 May 2020. During this period, COVID-19 cases exceeded 18,000 in Poland, with the study area showing prevalence at 31/100,000 and incidence at 6/100,000. The study included 6 hospitals with clinics or wards that diagnosed or hospitalized COVID-19 patients.

The study involved 441 healthcare workers. These employees were divided into two groups. The study group, defined as “frontline workers”, consisted of 206 healthcare workers (116 women and 90 men), working in places with the highest probability of contact with SARS-CoV-2, i.e., intensive care units, infectious diseases units, and emergency departments. The control group of “second-line workers” consisted of 235 healthcare workers (114 women and 121 men) working in wards other than the intensive care unit, infectious diseases unit, and emergency department. 

This study focused on the symptoms of anxiety, depression, and sleep disorders in all participants, using well-proven survey tools—the Generalized Anxiety Disorder scale (GAD-7; range 0–21) [18] to assess the severity of the symptoms of anxiety, the 9-item Patient Health Questionnaire (PHQ; range 0–27) [19,20,21,22,23,24] to assess the severity of depression symptoms, and the 7-item Insomnia Severity Index (ISI; range 0–28) to assess the severity of sleep disorders [25,26,27]. Based on the literature, we established cut-off points for the symptoms of anxiety, depression, and insomnia. Participants with scores below the cut-off points were characterized as without symptoms, whereas those with scores above the cut-off points were characterized as presenting symptoms.

Each participant reported basic demographic data, including gender (male or female), age, and workplace. Data on coexisting diseases such as hypertension (yes/no), diabetes mellitus (yes/no), coronary heart disease (yes/no), heart failure (yes/no), chronic obstructive pulmonary disease (yes/no), dyslipidemia (yes/no), asthma (yes/no), autoimmune diseases (yes/no), as well as cigarette smoking (yes/no) were also collected from each participant.

Before the study, we received the approval of the Ethics Committee of the Pomeranian Medical University (KB-0012/26/04/2020/Z) which conformed to the ethical guidelines of the Declaration of Helsinki. Each of the participants gave their informed written consent. Participants could withdraw from the survey at any time. The survey was anonymous and ensured the full confidentiality of information. 

### Statistical Analysis

A licensed Statistica 13.0 program (StatSoft, Tulsa, OK, USA) was used for statistical analysis. The assessment of normal distribution was performed using the Shapiro–Wilk test. The analysis of quantitative data was performed using Mann–Whitney’s U test. For the analysis of qualitative data, the *X*^2^ test was used; if the subgroup size was small, the Yates correction was applied. The evaluation of the relationship between the analyzed parameters was performed using univariable logistic regression model analysis and was corrected for potentially distorting data (age, gender, diagnosed hypertension, diabetes mellitus, dyslipidemia, asthma, autoimmune diseases, and cigarette smoking). Statistical significance was set at a *p* ≤ 0.05. 

## 3. Results

### 3.1. Comparison of Coexisting Diseases and Measurement Results between Frontline and Second-Line Medical Workers

The survey involved 441 healthcare professionals in Poland. There were 284 participants (64.4%) who presented symptoms of anxiety according to the GAD-7 score (>4 points), 312 participants (70.7%) presented depressive symptoms according to the PHQ-9 score (>4 points), and 256 participants presented symptoms of insomnia on the ISI scale (>8 points).

Frontline workers were significantly more likely to smoke cigarettes and were more likely to suffer from dyslipidemia compared to second-line workers (*p* = 0.025 and *p* = 0.010, respectively). This group, in comparison to the second-line workers, presented symptoms of anxiety, depression, and sleep disorders more often (*p* < 0.001, *p* < 0.001, *p* < 0.001, respectively). Frontline workers also significantly more often presented higher scores on all three scales (GAD-7, PHQ-9, and ISI) compared to second-line workers (*p* < 0.001, *p* < 0.001, *p* < 0.001, respectively). A case comparison is presented in Table 1.

### 3.2. Comparison of Health Factors between Frontline and Second-Line Healthcare Workers

Due to the low frequency of occurrence, the analysis did not take into account coronary heart diseases, heart failure, and chronic obstructive pulmonary disease.

The analysis of univariable logistic regression model showed that having the status of a frontline worker is associated with a much higher risk of intensifying symptoms of anxiety on the GAD-7 scale (OR = 1.340; *p* < 0.001), depression on the PHQ-9 scale (OR = 1.383; *p* < 0.001), and sleep disorders on the ISI scale (OR = 1.328; *p* < 0.001). The results are presented in Table 2. 

After the results were corrected for age, gender, and diagnosis of the following diseases: hypertension, diabetes mellitus, dyslipidemia, asthma, autoimmune diseases, and cigarette smoking, the increased risk of anxiety on the GAD-7 scale (OR = 1.934; *p* < 0.001), depression on the PHQ-9 scale (OR = 2.623; *p* < 0.001), and sleep disorders on the ISI scale (OR = 3.078; *p* < 0.001) were confirmed. The results are presented in Table 3. 

## 4. Discussion

The Centers for Disease Control and Prevention (CDC), based on the available information and clinical knowledge, has announced that the majority of SARS-CoV-2 infections are asymptomatic or oligosymptomatic, while the elderly and people of all ages with coexisting diseases may be at risk of severe COVID-19 [3,28]. Taking into account the above information, in this study, we have noted information on coexisting diseases such as hypertension, diabetes mellitus, coronary heart disease, heart failure, chronic obstructive pulmonary disease, dyslipidemia, asthma, autoimmune diseases, and nicotinism. 

Our analysis showed that healthcare workers working in places with the highest probability of contact with SARS-CoV-2, i.e., intensive care units, infectious disease units, or emergency departments, smoked cigarettes more frequently and were more likely to suffer from dyslipidemia compared to second-line workers (*p* = 0.025 and *p* = 0.010, respectively). 

Medical occupations are associated with hard-working conditions and an exceptional amount of stress. Daily contact with sick people, fatigue, stress, frequent lack of physical and even mental rest all increase the risk of error. All these factors are likely responsible for smoking among healthcare workers, whose levels may be even twice as high as in the general population, as shown in the study conducted by Ficarra et al. [29]. The results of our study show that the prevalence of smoking varies among medical professionals, confirming in some way the results of a study by Ruiz and Bayle, where hospital workers were more likely to smoke than primary healthcare workers [30]. The increased incidence of dyslipidemia observed among frontline workers in our study may have been directly associated with the more frequent tobacco smoking, a link confirmed in the general population by numerous studies [31,32,33,34]. 

In this study, a significant proportion of participants experienced symptoms of anxiety, depression, and insomnia, with over 90% prevalence of these symptoms in the group of employees having direct contact with persons suspected or infected with SARS-CoV-2, which differs from the frequency of these disorders in other studies [35,36,37]. This may be because our study was conducted in Europe where there has not been an infection problem on such a scale for nearly 100 years, while the above-mentioned studies were conducted on the Asian continent where epidemics are nothing exceptional. The health authorities in China, Hong-Kong, and Taiwan have well-prepared procedures, well-trained service personnel, and an industry that is well suited to these purposes.

Another observation in our study was that, compared to second-line workers, i.e., healthcare workers not directly involved in the diagnosis or therapy of patients infected with SARS-CoV-2, the group of frontline workers more often presented symptoms of anxiety, depression, and sleep disorders (*p* < 0.001, *p* < 0.001, *p* < 0.001, respectively). After the results were corrected for age, gender, and diagnosis of the following diseases: hypertension, diabetes, dyslipidemia, asthma, autoimmune diseases, and cigarette smoking, the increased risk of symptoms was confirmed among the frontline healthcare workers: anxiety on the GAD-7 scale (OR = 1.934; *p* < 0.001), depression on the PHQ-9 scale (OR = 2.623; *p* < 0.001), and sleep disorders on the ISI scale (OR = 3.078; *p* < 0.001) were confirmed. Similarly, Wu et al. emphasized that having the status of a healthcare worker working in high-risk areas, such as wards caring for SARS patients, was associated with 2–3 times higher levels of post-traumatic stress symptoms compared to people without such exposure [38].

A cross-sectional survey conducted in China among 1257 healthcare professionals in 34 hospitals showed that a significant proportion of those directly involved in the care of COVID-19 patients reported symptoms of depression (50.4%), anxiety (44.6%), insomnia (34%), and distress (71.5%). The status of a front-line worker was an independent risk factor for the deterioration of mental health outcomes [11]. Another Chinese cross-sectional observational study involving 180 health professionals providing direct care to COVID-19 patients showed that significant levels of anxiety and stress adversely affected the quality of sleep and their work [12]. 

Another observational study conducted in Singapore aimed to assess the prevalence of depression, stress, anxiety, and post-traumatic stress syndrome (PTSD) among all healthcare workers and to compare the results between medical and non-medical workers. Of the 470 participants, 14.5% showed symptoms of anxiety, 8.9% depression, 6.6% stress, and 7.7% PTSD [13]. These findings are in line with the findings of the present study.

In many countries, the SARS-CoV-2 pandemic has revealed serious problems in healthcare, including mental health. Healthcare workers working in emergency departments, infectious wards, and intensive care units are more likely to come into contact with infected people, and consequently their mental burden is significant. In addition, those workers who took part in our study were more likely to be affected by nicotinism and dyslipidemia, which, in relation to emerging media reports that co-morbidities can lead to a more complicated course of COVID-19, further exacerbates psychological problems, including intensified symptoms of anxiety, depression, insomnia, chronic fatigue, and stress. These people are exposed to emotionally challenging interactions with the sick and potentially dying person, they are worried about their own and their families’ health, they are subject to occupational overload due to staff shortages and insufficient personal protective equipment. It is precisely these workers in a state of mental decompensation that require reliable information support, stress reduction, and rest. In the case of many hours of continuous work, they should be guaranteed a place to relax on their own and to satisfy their daily needs such as food, sleep, protective clothing, and contact with their families [39]. Therefore, the mental health of healthcare workers who are in the first line of exposure to SARS-CoV-2 infection is not only an extremely important medical problem, but also a social one and it requires special attention.

Our study has several limitations. First of all, it lacks a longitudinal analysis of data. Secondly, the number of respondents is limited. Thirdly, it does not compare the symptoms of anxiety, depression, insomnia, and being a healthcare worker between various health centers in Poland from regions differing in the extent of the pandemic. This shows the need for longitudinal, multi-center studies with more respondents.

## 5. Conclusions

Our study showed that healthcare workers who are exposed to SARS-CoV-2-infected patients at emergency wards, infectious wards, and intensive care units are at a much higher risk of showing symptoms of anxiety, depression and sleep disorders than healthcare workers working in other wards. An extremely important public health task in the fight against the COVID-19 pandemic is to protect health professionals, especially those who fight on the frontline every day. 

## Figures and Tables

**Table 1 ijerph-17-05849-t001:** Comparison of selected parameters between frontline and second-line workers.

Characteristic		Frontline Workers (*n* = 206)	Second-Line Workers (*n* = 235)	*p*-Value
Sex (*n*, %)	women	116 (56.31%)	114 (48.51%)	0.102
	men	90 (43.69%)	121 (51.49%)	
Age [years], mean ± SD; Me		40.47 ± 4.93; 40.0	40.05 ± 5.51; 39.0	0.153
Do you have hypertension? (*n*, %)	no	184 (89.32%)	201 (85.53%)	0.233
	yes	22 (10.68%)	34 (14.47%)	
Do you have diabetes mellitus? (*n*, %)	no	203 (98.54%)	232 (98.72%)	0.803
	yes	3 (1.46%)	3 (1.28%)	
Do you have coronary heart disease? (*n*, %)	no	205 (99.51%)	235 (100.00%)	0.947
	yes	1 (0.49%)	0 (0.00%)	
Are you suffering from heart failure? (*n*, %)	no	206 (100.00%)	235 (100.00%)	1.000
	yes	0 (0.00%)	0 (0.00%)	
Do you have dyslipidemia? (*n*, %)	no	143 (69.42%)	188 (80.00%)	0.010
	yes	63 (30.58%)	47 (20.00%)	
Do you have asthma? (*n*, %)	no	194 (94.17%)	229 (97.45%)	0.136
	yes	12 (5.83%)	6 (2.55%)	
Do you have the chronic obstructive pulmonary disease? (*n*, %)	no	206 (100.00%)	234 (99.57%)	0.947
	yes	0 (0.00%)	1 (0.43%)	
Do you have autoimmune diseases? (*n*, %)	no	156 (75.73%)	191 (81.28%)	0.156
	yes	50 (24.27%)	44 (18.72%)	
Do you smoke cigarettes? (*n*, %)	no	91 (44.17%)	129 (54.89%)	0.025
	yes	115 (55.83%)	106 (45.11%)	
GAD-7 (*n*, %)	≤4	2 (0.97%)	155 (65.96%)	<0.001
	>4	204 (99.03%)	80 (34.04%)	
GAD-7, mean ± SD; Me		11.88 ± 3.93; 11.0	4.96 ± 5.12; 3.0	<0.001
PHQ-9 (*n*, %)	≤4	1 (0.49%)	128 (54.47%)	<0.001
	>4	205 (99.51%)	107 (45.53%)	
PHQ-9, mean ± SD; Me		14.19 ± 3.91; 13.5	6.00 ± 5,08; 4.0	<0.001
ISI (*n*, %)	≤8	0 (0.00%)	184 (78.30%)	<0.001
	>8	205 (100.00%)	51 (21.70%)	
ISI, mean ± SD; Me		17.14 ± 3.5; 16.0	7.26 ± 6.43; 5.0	<0.001

Abbreviations: GAD-7—Generalized Anxiety Disorder scale, PHQ-9—Patient Health Questionnaire, ISI—Insomnia Severity Index, *p*—statistical significance, *n*—number of patients, Me—median, SD—standard deviation.

**Table 2 ijerph-17-05849-t002:** Univariable logistic regression model among frontline and second-line healthcare workers.

Scale	Frontline Workers				Second-Line Workers			
	*p*-Value	OR	Cl −95%	Cl +95%	*p*-Value	OR	Cl −95%	Cl +95%
GAD7	<0.001	1.340	1.270	1.413	<0.001	0.746	0.707	0.787
PHQ-9	<0.001	1.383	1.309	1.461	<0.001	0.723	0.685	0.764
ISI	<0.001	1.328	1.265	1.393	<0.001	0.753	0.718	0.790

Abbreviations: GAD-7—Generalized Anxiety Disorder scale, PHQ-9—Patient Health Questionnaire, ISI—Insomnia Severity Index, *p*—statistical significance, OR—odds ratio, Cl—confidence interval.

**Table 3 ijerph-17-05849-t003:** Multivariable logistic regression model among frontline and second-line workers.

Scale	Frontline Workers				Second-Line Workers			
	*p*-Value	OR	Cl −95%	Cl +95%	*p*-Value	OR	Cl −95%	Cl +95%
GAD7	<0.001	1.934	1.726	2.167	<0.001	0.517	0.461	0.579
PHQ-9	<0.001	2.623	2.170	3.170	<0.001	0.381	0.316	0.461
ISI	<0.001	3.078	2.348	4.036	<0.001	0.325	0.248	0.426

Abbreviations: GAD-7—Generalized Anxiety Disorder scale, PHQ-9—Patient Health Questionnaire, ISI—Insomnia Severity Index, *p*—statistical significance, OR—odds ratio, Cl—confidence interval.

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
