# Peer review of "Assessment of Mental Health Factors among Health Professionals Depending on Their Contact with COVID-19 Patients"

_ijerph, 2020, doi:10.3390/ijerph17165849_

Round 1

Reviewer 1 Report

In this manuscript Wankovic et al aim to evaluate the mental status of health workers in contact with COVID pandemic in a región in Poland. Additionally, the authors investigate posible co-morbidities in the study subjects. This assessment has been performed by comparing with a group of health profesional not directly in contact with COVID patients. In addtion to the limitations described by the authors, another bias could decrease the extension of the conclusions reached. This limitation is related to the fact that stress, anxiety or depression are subjective variables and those professionals dealing with COVID could tend to exagerate their answers in the questionnaire as they may feel neglected by the health authorities and may need more personnel. The comparison study of hypertension, diabetes or dyslipemia is objective and, in my opinion, is the main contribution of the study. To give support to this line the authors should argue in the discussion if the blood tests performed are suitable to asesses the status of these diseases given the period of time, not very long, of the pandemic. Detailed information about the number of individuals subjected to blood tests and when these tests were carried out is needed.     

Minor points,

The prevalence and incidence of COVID-19 in the area of study during the period of study should be discussed.

Intro. The COVID-19 outbreak was decleared in China in December 2019 not at the beginnig of 2020.

More abundant reference of previous works dealing with the subject of this study should be commented in the discussion.

Author Response

Thank you very much for your valuable comments and advice. We have made changes in accordance with them.

Point 1: The prevalence and incidence of COVID-19 in the area of study during the period of study should be discussed.

Response 1: 72 This is a cross-sectional, hospital-based study conducted in the Western Pomerania region in Poland from 3 May 2020 to 17 May 2020. During this period, COVID-19 cases exceeded 18,000 in Poland, with the study area showing prevalence at 31/100,000 and incidence at 6/100,000 74 .

Point 2: Intro. The COVID-19 outbreak was decleared in China in December 2019 not at the beginnig of 2020.

Response 2: 28 In December 2019, a new type of coronavirus was identified in Wuhan.

Point 3: More abundant reference of previous works dealing with the subject of this study should be commented in the discussion.

Response 3: 209 A cross-sectional survey conducted in China among 1257 healthcare professionals in 34 hospitals showed that a significant proportion of those directly involved in the care of COVID-19 patients reported symptoms of depression (50.4%), anxiety (44.6%), insomnia (34%) and distress (71.5%). The status of a front-line worker was an independent risk factor for the deterioration of mental health outcomes [11]. Another Chinese cross-sectional observational study involving 180 health professionals providing direct care to COVID-19 patients showed that significant levels of anxiety and stress adversely affected the quality of sleep and their work [12]. Another observational study conducted in Singapore aimed to assess the prevalence of depression, stress, anxiety and post-traumatic stress syndrome (PTSD) among all healthcare workers and to compare results between medical and non-medical workers. Of the 470 participants, 14.5% showed symptoms of anxiety, 8.9% depression, 6.6% stress and 7.7% PTSD [13]. These findings are in line with the findings of the present study 227.

Reviewer 2 Report

Author should be revised following:

Introduction

- Author should be added the number case in your country and impact on health, social, economic views.

- Author should be stated the mental health problem from COVID-19 in the world and your country.

-Author should be added the detail of your sample area or setting.

- Author should be added the update publication related with your study.

Methodology

- Author should be added the detail of calculation for your sample size.

- Author should be added the sampling frame of your sample size.

- Author should be added the detail process of pre-test measurement.   

- Author should be added the detail of data cut of point in each parameter.

- Author should be added the detail of qualitative data procedure.

Discussion

- Please concern this paragraph “ While searching available medical databases, we did not find any other study that compares 123 mental health factors among medical personnel in the times of the SARS-CoV-2 pandemic while 124 taking into account coexisting diseases”.

- Please concern this paragraph  “Our study has several limitations. First of all, it lacks a longitudinal analysis of data. Secondly, 182 the number of respondents is limited. Thirdly, it does not compare the symptoms of anxiety, 183 depression, insomnia, and being a health care worker between various health centers in Poland from 184 regions differing in the extent of the pandemic. This shows the need for longitudinal, multi-center 185 studies with more respondents.”

Conclusion

  • Author should be added more your highlight finding with the strong recommendation

Author Response

Thank you very much for your valuable comments and advice. According to them, we have introduced changes within the limits that do not disturb the narrative of the article.

 Point 1: Author should be added the number case in your country and impact on health, social, economic views. Author should be stated the mental health problem from COVID-19 in the world and your country.

Response 1: 33 In Poland, there have been 45000 confirmed cases and 1700 deaths. 39 The lockdown introduced by the Polish government on March 15, 2020 dramatically affected the daily life of the society. Isolation and many other imposed limitations may increase psychological distress, including the symptoms of depression and anxiety, as confirmed by research conducted around the world. Although this response of the human psyche to threats and uncertainty is natural, in some individuals it may exceed an ability to adapt and cope, leading to the development of clinically significant symptoms. Certain groups of people are particularly vulnerable to stress due to the consequences of the pandemic and, as a result, are more likely to develop depression and generalised anxiety disorder. For some, this may be related problems from before the epidemic, e.g. people with a precarious financial situation may fear that the pandemic will make it even worse; those entering adulthood and still living with their parents may face increased interpersonal conflicts at home. For others, previously natural situations, may develop into difficult situations, increasing anxiety or depression, such as when living alone may lead to the overwhelming sense of loneliness when meeting other becomes impossible. Others have to deal with completely new and serious challenges, e.g. students and teachers who switch to online education and fear of not being able to cope, with insufficient resources and competence in the field of new communication technologies. However, the group of people who seem to be most exposed to exceptionally high level of stress during the pandemic are health professionals 59.

Point 2: Author should be added the detail of your sample area or setting.

Response 2: 72 This is a cross-sectional, hospital-based study conducted in the Western Pomerania region in Poland from 3 May 2020 to 17 May 2020. During this period, COVID-19 cases exceeded 18,000 in Poland, with the study area showing prevalence at 31/100,000 and incidence at 6/100,000. The study included 6 hospitals with clinics or wards that diagnosed or hospitalized COVID-19 patients 75

Point 3: Author should be added the update publication related with your study.

Response 3: 209 A cross-sectional survey conducted in China among 1257 healthcare professionals in 34 hospitals showed that a significant proportion of those directly involved in the care of COVID-19 patients reported symptoms of depression (50.4%), anxiety (44.6%), insomnia (34%) and distress (71.5%). The status of a front-line worker was an independent risk factor for the deterioration of mental health outcomes. Another Chinese cross-sectional observational study involving 180 health professionals providing direct care to COVID-19 patients showed that significant levels of anxiety and stress adversely affected the quality of sleep and their work. Another observational study conducted in Singapore aimed to assess the prevalence of depression, stress, anxiety and post-traumatic stress syndrome (PTSD) among all healthcare workers and to compare results between medical and non-medical workers. Of the 470 participants, 14.5% showed symptoms of anxiety, 8.9% depression, 6.6% stress and 7.7% PTSD. These findings are in line with the findings of the present study 227.

Point 4: Author should be added the detail of calculation for your sample size.

Response 4: We calculated the sample size in the assessment of selected index for the chi square test with the test power of 0.9.

Point 5:Author should be added the sampling frame of your sample size.

Response 5: The study included 6 hospitals with clinics or wards that diagnosed or hospitalized COVID-19 patients in the Western Pomerania region in Poland. The study involved 441 health care workers. These employees were divided into two groups. The study group, defined as “frontline workers”, consisted of 206 health care workers (116 women and 90 men), working in places with the highest probability of contact with SARS-CoV-2, i.e. intensive care units, infectious diseases units, and emergency departments. The control group of "second-line workers" consisted of 235 health care workers (114 women and 121 men) working in wards other than the intensive care unit, infectious diseases unit, and emergency department. Each participant reported basic demographic data, including gender (male or female), age, and workplace. Data on coexisting diseases such as hypertension (yes/no), diabetes mellitus (yes/no), coronary heart disease (yes/no), heart failure (yes/no), chronic obstructive pulmonary disease (yes/no), dyslipidemia (yes/no), asthma (yes/no), autoimmune diseases (yes/no), as well as cigarette smoking (yes/no) were also collected from each participant. Each of the participants gave their informed written consent. Participants could withdraw from the survey at any time. The survey was anonymous and ensured the full confidentiality of information.

Point 6:Author should be added the detail process of pre-test measurement.   

Response 6: At the stage of planning the research, in the evaluation of selected indicators, it was set that the statistical significance was <0.05 and the test power was 0.90. On this basis, it was calculated that a minimum of 203 people would be needed in each of the studied groups. In our study, we tested 206 people in the first group and 235 in the second group. We obtained much higher levels of statistical significance.

Point 7:Author should be added the detail of data cut of point in each parameter.

Response 7:  The cut off point in each parameter was made on the basis of literature analysis. In our opinion therefore cut off point is optimal.

Point 8:Author should be added the detail of qualitative data procedure.

Response 8:   The qualitative data are presented as number and a percentages. Statistical significance levels were calculated from the X2 test if the subgroup size was small, the Yates correction was applied.

Point 9: Please concern this paragraph “ While searching available medical databases, we did not find any other study that compares 123 mental health factors among medical personnel in the times of the SARS-CoV-2 pandemic while 124 taking into account coexisting diseases”.

Response 9:   Due to a lack of understanding, this fragment has been removed.

Point 10: Please concern this paragraph  “Our study has several limitations. First of all, it lacks a longitudinal analysis of data. Secondly, 182 the number of respondents is limited. Thirdly, it does not compare the symptoms of anxiety, 183 depression, insomnia, and being a health care worker between various health centers in Poland from 184 regions differing in the extent of the pandemic. This shows the need for longitudinal, multi-center 185 studies with more respondents.”

Response 10:   The manuscript mentions in this place about the limits of our study.

Point 11: Author should be added more your highlight finding with the strong recommendation

Response 11: 255 An extremely important public health task in the fight against the COVID-19 pandemic is to protect health professionals, especially those who fight on the frontline every day 256.

Round 2

Reviewer 2 Report

This revision manuscript is accepted for publication.